# Detection of SARS-CoV-2 and Other Infectious Agents in Lower Respiratory Tract Samples Belonging to Patients Admitted to Intensive Care Units of a Tertiary-Care Hospital, Located in an Epidemic Area, during the Italian Lockdown

**DOI:** 10.3390/microorganisms9010185

**Published:** 2021-01-16

**Authors:** Adriana Calderaro, Mirko Buttrini, Sara Montecchini, Giovanna Piccolo, Monica Martinelli, Maria Loretana Dell’Anna, Alan Di Maio, Maria Cristina Arcangeletti, Clara Maccari, Flora De Conto, Carlo Chezzi

**Affiliations:** 1Department of Medicine and Surgery, University of Parma, Viale A. Gramsci 14, 43126 Parma, Italy; mirko.buttrini@unipr.it (M.B.); giovanna.piccolo@unipr.it (G.P.); mariacristina.arcangeletti@unipr.it (M.C.A.); clara.maccari@unipr.it (C.M.); flora.deconto@unipr.it (F.D.C.); carlo.chezzi@unipr.it (C.C.); 2Unit of Clinical Virology, University Hospital of Parma, Viale A. Gramsci 14, 43126 Parma, Italy; smontecchini@ao.pr.it (S.M.); mdellanna@ao.pr.it (M.L.D.); 3Unit of Clinical Microbiology, University Hospital of Parma, Viale A. Gramsci 14, 43126 Parma, Italy; mmartinelli@ao.pr.it (M.M.); adimaio@ao.pr.it (A.D.M.)

**Keywords:** respiratory viruses, SARS-CoV-2, diagnosis, bacteria, pandemic, molecular assays, fungi

## Abstract

The aim of this study was the detection of infectious agents from lower respiratory tract (LRT) samples in order to describe their distribution in patients with severe acute respiratory failure and hospitalized in intensive care units (ICU) in an Italian tertiary-care hospital. LRT samples from 154 patients admitted to ICU from 27 February to 10 May 2020 were prospectively examined for respiratory viruses, including SARS-CoV-2, bacteria and/or fungi. SARS-CoV-2 was revealed in 90 patients (58.4%, 72 males, mean age 65 years). No significant difference was observed between SARS-CoV-2 positives and SARS-CoV-2 negatives with regard to sex, age and bacterial and/or fungal infections. Nonetheless, fungi were more frequently detected among SARS-CoV-2 positives (44/54, 81.4%, *p* = 0.0053). *Candida albicans* was the overall most frequently isolated agent, followed by *Enterococcus faecalis* among SARS-CoV-2 positives and *Staphylococcus aureus* among SARS-CoV-2 negatives. Overall mortality rate was 40.4%, accounting for 53 deaths: 37 among SARS-CoV-2 positives (mean age 69 years) and 16 among SARS-CoV-2 negatives (mean age 63 years). This study highlights the different patterns of infectious agents between the two patient categories: fungi were prevalently involved among SARS-CoV-2-positive patients and bacteria among the SARS-CoV-2-negative patients. The different therapies and the length of the ICU stay could have influenced these different patterns of infectious agents.

## 1. Introduction

The novel coronavirus SARS-CoV-2 was reported for the first time in China, in Hubei province, in December 2019 and has rapidly spread around the world [1,2,3]. As of 10 May 2020, there have been 3,917,366 confirmed cases of the novel coronavirus disease globally and 274,361 deaths [4]. Most of the cases (1,245,775) and deaths (75,364) were reported in the U.S.A., followed by Spain (223,578 cases and 26,478 deaths), Italy (218,268 cases and 30,395 deaths), the United Kingdom (215,264 cases and 31,587 deaths), the Russian Federation (209,688 cases and 1915 deaths), Germany (169,218 cases and 7395 deaths), Turkey (137,115 cases and 3739 deaths) and France (137,008 cases and 26,268 deaths) [4]. At the time of submission of this manuscript, there have been, globally, 70,476,836 confirmed cases and 1,599,922 deaths; in particular, in Italy, 1,825,775 confirmed cases with 64,036 deaths have been reported [5].

SARS-CoV-2 has a high infection rate with the estimates of the reproductive number (R0) ranging from 1.4 to 2.5, according to World Health Organization (WHO) [6], and increasing up to 2.2–5.7, as observed in other studies [7,8,9,10]. Studies conducted on the new coronavirus estimated the mean incubation period to be 6.4 days (2.1–11.1 days) [11]. The incubation period for SARS-CoV-2 is comparable to other recent epidemic respiratory viral diseases, such as severe acute respiratory syndrome, or SARS (2–7 days) [12], and Middle East respiratory syndrome, or MERS (2–14 days) [13], whereas it is slightly longer than that of swine flu (1–4 days) and seasonal influenza (1–4 days) [14].

In Italy, the first case of SARS-CoV-2 infection was assessed on 7 February 2020 in a non-Italian subject recently coming from China, but the infection has become epidemic since 21 February 2020, when autochthonous cases were identified in Northern Italy [15]. In the area of Parma, where our laboratory is located, the first case was reported from a suckling baby whose sample was sent to the laboratory on 25 February 2020 [16], and SARS-CoV-2 became epidemic on 3 March 2020, with a total of 3298 cases and 710 deaths recorded until 10 May 2020 [17].

The aim of this study was to detect SARS-CoV-2 and other infectious agents from the lower respiratory tract samples in order to describe their distribution in patients with severe acute respiratory failure hospitalized in intensive care units (ICU) in a tertiary-care hospital, located in the epidemic area of Northern Italy, during the Italian SARS-CoV-2 lockdown [17].

## 2. Materials and Methods

### 2.1. Samples and Patients

A total of 301 samples (299 bronchial aspirates (BAS) and 2 bronchoalveolar lavages (BAL)) sent to our laboratory for viral, bacterial and/or fungal diagnostic purposes belonging to 154 patients (116 males and 38 females, mean age 60 years, ranging from 9 days to 83 years) admitted to the ICU of the University Hospital of Parma (Italy) in the period 27 February (when the first respiratory sample of a COVID-19 suspected case admitted in ICU was sent to the laboratory)—10 May 2020 were prospectively examined. For a subset of 91 patients, COVID-19 positivity was reported in the clinical data present in the medical order of the first sample. Multiple samples were sent to our laboratory from 80 patients during the ICU stay (2 samples from 36 patients, 3 samples from 28 patients, 4 samples from 9 patients and 5 samples from 7 patients). From the remaining 74 patients, only one sample was sent to our laboratory. 

For 129 patients for whom the date of admission to ICU was available, the average time between the admission and the first sample collection for the laboratory diagnosis of respiratory tract infection, hereafter named time-to-collection (TTC), was 11 days, ranging from 0 to 86 days. 

The mortality rate was calculated for 131 patients (99 males and 32 females) for whom the information was available during the study period.

Laboratory diagnosis was performed, and a clinical report was produced upon medical request for a clinical suspicion of infection of the lower respiratory tract. Anonymization of patients was conducted before data analysis, and medical information necessary to produce the laboratory diagnosis was protected and used in aggregate form. The samples analyzed in this study were sent to the University Hospital of Parma for diagnostic purposes, and the laboratory diagnosis results were reported in the medical records of the patients as an answer to a clinical suspicion indicated in the medical order; ethical approval at the University Hospital of Parma is required only in cases in which the clinical samples are to be used for applications other than diagnosis.

### 2.2. Definitions

A patient was defined as SARS-CoV-2-positive when at least one lower respiratory tract sample was SARS-CoV-2 RNA-positive. When SARS-CoV-2 positivity was not obtained by our molecular method but rather from the information reported in the medical order, the results were analyzed separately.

### 2.3. Respiratory Virus Detection and Identification

The qualitative detection of SARS-CoV-2 RNA was carried out for all samples by using a real-time PCR after reverse transcription (RT-PCR) panel according to the Centers for Disease Control and Prevention (CDC, Atlanta, GA, USA) protocol, with some modifications as previously reported [18].

An aliquot of the first sample of each patient was used to detect the nucleic acids of adenovirus, influenza A virus H1 and H3, influenza B virus, parainfluenza virus types 1–4, respiratory syncytial virus types A and B, metapneumovirus, bocavirus types 1–4, rhinovirus types A, B, and C, enterovirus and coronavirus types 229E, NL63 and OC43, with the one-step real-time reverse transcription PCR Allplex™ Respiratory Panels 1–3 assays (Seegene Inc., Seoul, Korea), as described previously [18,19]. In parallel, these samples were also submitted to a rapid cell culture method in 8-well chamber slides (Thermo Fisher Scientific, Monza, Italy) for the detection of influenza A and B viruses, parainfluenza 1-3 viruses and respiratory syncytial virus by indirect immunofluorescence assay (IIF) with monoclonal antibodies (Thermo Fisher Scientific), as previously described [19]. Moreover, an aliquot of the same samples was also submitted to a conventional cell culture method; when a cytopathic effect was observed, the identification was performed as previously described [16,18,19]

### 2.4. Bacterial and Fungal Detection and Identification

All samples were inoculated onto blood agar and MacConkey agar Petri dishes (KIMA, Padova, Italy) and incubated for 24 h at 37 °C. After pure microbial growth was achieved (24–72 h), identification tests were performed by matrix-assisted laser desorption/ionization time-of-flight (MALDI-TOF) mass spectrometry using a VITEK MS instrument (bioMérieux, Marcy-l’Étoile, France) for the identification of Gram-positive bacteria and Autoflex Speed—MALDI Biotyper mass spectrometer (Bruker Daltonics, Bremen, Germany) in the case of Gram-negative bacteria, according to the laboratory workflow [16,20], and following the manufacturer’s instructions.

An aliquot of the first respiratory sample of each patient was used to detect the nucleic acids of *Chlamydophila pneumoniae*, *Mycoplasma pneumoniae*, *Legionella pneumophila*, *Bordetella pertussis*, *Bordetella parapertussis*, *Streptococcus pneumoniae* and Haemophilus influenzae by the Allplex™ Respiratory Panel 4 real-time PCR assay (Seegene Inc.), according to the manufacturer’s instructions.

For fungal detection, when requested on the medical order or a growth of fungal colonies was observed from bacterial culture, a total of 169 samples were also inoculated onto Sabouraud dextrose agar with chloramphenicol (KIMA) and incubated at 30 °C in the appropriate conditions [20,21]. After adequate (at least two to three colonies) pure microbial growth was achieved, the identification was performed by a VITEK MS mass spectrometer (bioMérieux) or conventional methods, according to the laboratory workflow [20,21].

### 2.5. Statistical Analysis

Yates’s chi-square test was used for comparison of the different categories considered. In addition, the odds ratio (OR) was calculated in order to evaluate the strength of the associations that emerged. Student’s *t*-test was used to compare the means of different values such as the age, the TTC and the number of agents between SARS-CoV-2-positive and SARS-CoV-2-negative patients also in relation to the mortality. Statistical significance was set at *p* < 0.05.

## 3. Results

### 3.1. SARS-CoV-2 RNA Detection

The RT-PCR for the detection of SARS-CoV-2 RNA performed on the 301 samples (accounting for 154 patients) gave a positive result in 155 samples (51.5%). Among the overall 154 patients, 90 patients (58.4%: 72 males and 18 females, mean age 60 years, range 32–76 years) were SARS-CoV-2-positive: 87 on the first tested sample (96.7%) and 3 on the second one (on average, 7 days after the first, ranging from 4 to 9 days). The remaining 64 patients (41.6%: 44 males and 20 females, mean age 62 years, range 9 days–83 years) were SARS-CoV-2-negative on all 84 tested samples. No difference was observed between the SARS-CoV-2-positive and the SARS-CoV-2-negative patients with regard to sex (*p* = 0.16; OR: 1.82) and the mean age (*p* = 0.2775). 

With regard to the temporal distribution, a peak of SARS-CoV-2-positive cases in ICU was observed between March 19 and April 6, when a total of 51 new positive cases were detected (Figure 1).

The average TTC was 8 days (ranging from 0 to 34 days) for the 86 SARS-CoV-2-positive patients (for whom the TTC was available considering the first SARS-CoV-2-positive sample) and 15 days (ranging from 0 to 86 days) for the 43 SARS-CoV-2-negative patients (for whom the TTC was available considering the first sample examined) (*p* = 0.0009).

Among the 63 SARS-CoV-2-positive patients with at least one examined sample besides the first positive one, 26 (41.3%) were still SARS-CoV-2 RNA-positive on the last sample (on average, 11 days after the first positive one, range 4–28 days), including two cases (with three samples each) with an in-between negative result, and 37 (58.7%) became SARS-CoV-2 RNA-negative (on average, 17 days from the first, range 5–39 days), including one case (with five examined samples) with an 18-days negative result followed by a positive one at 26 days from the first positive sample before becoming negative on both the 36- and 46-days samples. 

Among 131 patients (99 males and 32 females), the mortality rate was 40.4%, accounting for 53 deaths. In particular, 37 deaths (30 males and 7 females, accounting for 44.1% and 41.1% of the 68 positive males and 17 females, respectively) occurred among SARS-CoV-2-positive patients (37/85, 43.5%) and 16 (10 males and 6 females, accounting for 32.2% and 40% of the 31 negative males and 15 females, respectively) among SARS-CoV-2-negative patients (16/46, 34.7%). Among the 37 SARS-CoV-2-positive dead patients, 18 had multiple samples: in 11 cases (61.1%), at least one SARS-CoV-2-negative sample was found (on average, 16 days after SARS-CoV-2 RNA detection), while in 7 cases (38.9%), all examined samples were positive. 

No difference between the SARS-CoV-2-positive and the SARS-CoV-2-negative patients was observed considering the number (*p* = 0.4312, OR= 1.45), the sex (*p* = 0.2732, OR = 2.57) and the mean age (65 years vs. 69 years; *p* = 0.054) of the dead patients. However, among SARS-CoV-2-positive patients, the dead patients were significantly older (65 years) than those who survived (56 years) (*p* < 0.00001), whereas no difference in the mean age between the dead (69 years) and survived (63 years) SARS-CoV-2-negative patients was found (*p* = 0.08). 

### 3.2. Detection of Infectious Agents Other than SARS-CoV-2

The detection of other infectious agents revealed that, among the 90 SARS-CoV-2-positive patients (accounting for 217 examined samples, including 127 for which also fungi were searched for), 77 (85.5%) had a bacterial and/or fungal co-infection and 13 (14.5%) had no co-infection, whereas among the 64 SARS-CoV-2-negative patients (accounting for 84 examined samples, including 42 for which also fungi were searched for), 48 (75%) had a bacterial and/or fungal infection and 16 (25%) had no infection (Table 1). No virus other than SARS-CoV-2 was detected.

When only the first sample was considered, except in three cases for which the second sample (SARS-CoV-2-positive) was included, instead of the first one (SARS-CoV-2-negative), the difference observed in the presence of bacteria and fungi between SARS-CoV-2-positive and SARS-CoV-2-negative patients (54 vs. 45 patients with bacterial and/or fungal detection, respectively) was not statistically significant (*p* = 0.05; OR: 2.07) (Table 2). However, among patients with bacterial and/or fungal infections, a fungal infection in the first sample was more frequently detected among the SARS-CoV-2-positive patients (44/54, 81.4%) than the SARS-CoV-2-negative patients (24/45, 53.3%) (*p* = 0.0053, OR: 3.85). 

In particular, among the SARS-CoV-2-positive patients, a total of 75 agents other than SARS-CoV-2 were found: 28 bacteria (Figure 2A) and 47 fungi (Figure 2B). *Enterococcus faecalis* (11) and *Klebsiella pneumoniae* (4) were the most frequently isolated bacteria, while *Candida albicans* (36) was the most frequently isolated agent among yeasts. Among the SARS-CoV-2-negative patients, a total of 55 bacteria (Figure 2A) and 26 fungi were found (Figure 2B). *Staphylococcus aureus* (13) and *Pseudomonas aeuginosa* (7) were the most frequently isolated bacteria, while *Candida albicans* (17) and *Candida glabrata* (4) represented the most frequently isolated fungi. The detailed list of the different agents found for each sample of each SARS-CoV-2-positive and SARS-CoV-2-negative patient is reported in Appendix A, respectively.

The average number of agents involved in co-infection among the SARS-CoV-2-positive patients was significantly lower (0.8 agents for each patient) than that observed among SARS-CoV-2-negative patients (1.3 agents for each patient) (*p* = 0.0093).

In 22 out of the 37 SARS-CoV-2-positive dead patients (59.5%), a co-infection with bacteria (2 patients, 5.4%), fungi (16 patients, 43.2%) or both (4 patients, 10.8%) was found; in the remaining 15 patients (40.6%), no co-infection was found.

Among the 16 SARS-CoV-2-negative dead patients, for 13 (81.3%) of them, an infection with bacteria (5 patients, 31.3%), fungi (4 patients, 25%) or both (4 patients, 25%) was found; no infection was found in the remaining three patients (18.7%). No statistically significant difference was observed in the number of dead patients between SARS-CoV-2 positives and SARS-CoV-2 negatives with regard to the presence of bacterial and/or fungal infections (*p* = 0.2218; OR: 0.34).

For 40 out of the 48 SARS-CoV-2-positive survived patients, two or more samples were available during the study period: 23 patients became SARS-CoV-2 RNA-negative (on average, 17 days after the first positive sample) and 17 remained positive. For the remaining eight patients, only one sample was examined. In 30 out of the 48 SARS-CoV-2-positive survived patients (62.5%), a co-infection with bacteria (7 patients, 14.6%), fungi (15 patients, 31.3%) or both (8 patients, 16.7%) was found; in the remaining 18 patients (37.5%), no co-infection was observed.

Among the 30 SARS-CoV-2-negative survived patients (62.5%, 30 out of the 46 SARS-CoV-2-negative patients for whom the clinical data were available), an infection was found in 23 cases (76.6%): 13 with bacteria (43.3%), 4 with fungi (13.3%) and 6 with either bacteria or fungi (20%). For the remaining seven patients (23.4%), no infection was found.

With regard to survived patients, no statistically significant difference was observed between SARS-CoV-2-positive patients with bacteria and/or fungi co-infection (30 out of 48 patients) and SARS-CoV-2-negative patients with bacteria and/or fungi infection (23 of 30 patients) (*p* = 0.29144; OR: 0.51).

### 3.3. Adjustment by COVID-19 Positivity from Medical Order

In 76 out of the 91 patients for whom COVID-19 positivity was reported on the medical order, the detection of SARS-CoV-2 RNA on lower respiratory tract samples gave a positive result, whereas in 15 patients, it was negative. Among these latter patients, the mean TTC (calculated for 14 patients, for which the time of admission was available) was 20 days (range 0–39 days). If the mortality rate was adjusted by considering these 15 patients (9 dead and 6 survived) as SARS-CoV-2-positive, the difference of the deaths between the SARS-CoV-2-positive (46 out of 105, 43.8%) and the SARS-CoV-2-negative patients (7 out of 49, 14.3%) became statistically significant (*p* = 0.0007, OR: 4.68). Similarly, the mean age of SARS-CoV-2-positive dead patients (65 years, range from 32 to 81 years) was statistically lower than that of the SARS-CoV-2-negative dead patients (74 years) (*p* = 0.0056). In particular, among SARS-CoV-2-positive patients, 57% of those aged 60–70 years and 100% of those over 70 years died.

## 4. Discussion

COVID-19 is a severe global health issue caused by SARS-CoV-2, which results in respiratory illness, such as SARS-CoV and MERS-CoV, and may cause death in severe cases. The mortality is significantly higher in the elderly age group, mostly having pre-existing disease conditions [22]. Severe cases require intubation and mechanical ventilation [23]. This study, through the detection of SARS-CoV-2 and other infectious agents from lower respiratory tract samples, describes their distribution in patients with severe acute respiratory failure admitted to the ICU of the University Hospital of Parma, a tertiary-care hospital located in the epidemic area of Northern Italy, in a 3-month period (27 February–10 May), during the Italian 2020 SARS-CoV-2 lockdown [17].

The data at hand show that, in the period in which our country carried out the lockdown (9 March–18 May 2020) due to the pandemic, 58.4% of the patients admitted to the ICU of our hospital and submitted to laboratory diagnosis for respiratory infection were SARS-CoV-2-positive. The mean TTC could have influenced the attribution of the patient category (SARS-CoV-2-positive or -negative). As a matter of fact, the mean TTC among the SARS-CoV-2-positive patients (limited to those for whom the time of admission was known) was significantly shorter (8 days) than that observed among the SARS-CoV-2-negative patients (15 days), including those patients for whom a diagnosis of COVID-19 was available from the medical order. The SARS-CoV-2 RNA-negative results obtained from these latter patients were likely due to the long mean TTC (20 days).

Based on the results of the present study, 58.7% of the SARS-CoV-2-positive patients became negative during hospitalization in ICU, whereas for 41.3%, even the last examined sample was SARS-CoV-2-positive. This could be due to the fact that, among these latter patients, the average time between the first and the last sample examined was shorter (11 days) than that required for the viral clearance (on average, 17 days from the first positive sample), as observed among the patients for whom a SARS-CoV-2-negative result was achieved.

In this study, limited to the patients for whom the information was available, the mortality rate of SARS-CoV-2-positive patients admitted in ICU (43.5%, 37/85) was higher than that observed among the SARS-CoV-2-negative patients (34.7%, 16/46). The mortality rate of SARS-CoV-2-positive patients would further increase to 43.8% (46/105) if the 15 patients with COVID-19 positivity reported in the medical order were included in such group, becoming statistically significant in comparison to that found among SARS-CoV-2-negative patients (22.6%, 7/31). Our higher mortality rate, in comparison to that reported by Grasselli et al. [23] in a 1-month study (20 February to 18 March 2020) in Lombardy, the Italian region most affected by the pandemic, could be due to the longer period of the present study which included the peak (from March 19 to April 6, overlapping the Italian epidemic one 21–28 March) of SARS-CoV-2-positive patients admitted in ICU in our area located close to the border of the region of Lombardy.

With regard to the detection of infectious agents other than SARS-CoV-2 among the overall examined samples, 85.5% of the SARS-CoV-2 positives and 75% of the SARS-CoV-2 negatives showed the presence of bacteria and/or fungi, suggesting that SARS-CoV-2-positive patients could have more frequent co-infections. However, these data could have been affected by the different number of multiple samples examined for the two patient categories, likely due to a different length of stay. For this reason, the comparison of the detection of bacterial and/or fungal infections between the two different patient categories was performed taking into account only the first sample. Although no significant difference between the number of SARS-CoV-2-positive patients with co-infection and that of SARS-CoV-2-negative patients with infection was observed (even if very close to the limit, *p* = 0.05), the number of agents involved in co-infections was lower among SARS-CoV-2 positives than among the SARS-CoV-2 negatives. Nonetheless, a fungal infection was more frequently detected among the SARS-CoV-2-positive patients (81.4%) than the SARS-CoV-2-negative patients (53.3%), likely due to the strong therapeutic pressure. Limited to the patients for whom the information on the use of the antibiotic therapy at the time of the first sample collection was available, the SARS-CoV-2 positives were prevalently treated with azithromycin and ceftriaxone mainly in combination (25 cases) followed by piperacillin-tazobactam (18 cases). Among the SARS-CoV-2 negatives, azithromycin was never indicated either in combination or alone, and several different antibiotics, alone or in combination, mainly involving penicillin with the β-lactamase inhibitor (14 cases) and carbapenems (9 cases) were reported. Antifungal therapy was in use in one SARS-CoV-2-positive patient and in four SARS-CoV-2 negatives.

Overall, *Candida albicans* was the most frequently detected agent (36 among SARS-CoV-2 positives and 17 among SARS-CoV-2 negatives). Among bacteria, a different distribution of agents was identified between the two patient categories: *E. faecalis* was predominant among the SARS-CoV-2-positive patients, whereas *S. aureus* was the most frequent bacteria found among SARS-CoV-2-negative patients.

No co-infection between SARS-CoV-2 and other respiratory viruses was found. This result is in agreement with those reported in the same area in a different period (1 December 2019–31 March 2020) in which among in- and out-patients, viral mixed infections were observed only in three children younger than 1 year old when samples of the upper respiratory tract (two nasopharyngeal swabs and one throat swab) were analyzed [18].

In our study, SARS-CoV-2 positivity mainly involved males rather than females (72 vs. 18) but no statistical difference emerged, unlikely reported by other authors [23,24]. The average age (65 years) of the SARS-CoV-2-positive dead patients (range from 32 to 81 years) was lower than that of SARS-CoV-2-negative patients (74 years), as also reported in the literature in Italy in the same study period [25,26]. On the contrary, among SARS-CoV-2-positive patients, the older patients more frequently died, likely due to the fact that more often they have co-morbidities [27].

## 5. Conclusions

The data reported in this study suggest that SARS-CoV-2-positive patients admitted to ICU are prevalently men, aged from 50 to 70 years, and that the higher mortality rate observed among SARS-CoV-2-positive patients, in comparison to SARS-CoV-2-negative patients, is mainly attributable to older patients regardless of sex.

This study highlights the different patterns in the distribution of infectious agents between the two patient categories: fungi were prevalently involved among SARS-CoV-2-positive patients and bacteria among the SARS-CoV-2-negative patients. Moreover, between the two patient categories, also a different distribution of bacteria was observed (*E. faecalis* vs. *S. aureus*). The different therapies and the length of the ICU stay could have influenced these different patterns of infectious agents. However, the role of fungal and bacterial infections in the COVID-19 disease course requires further investigation.

## Figures and Tables

**Figure 1 microorganisms-09-00185-f001:**
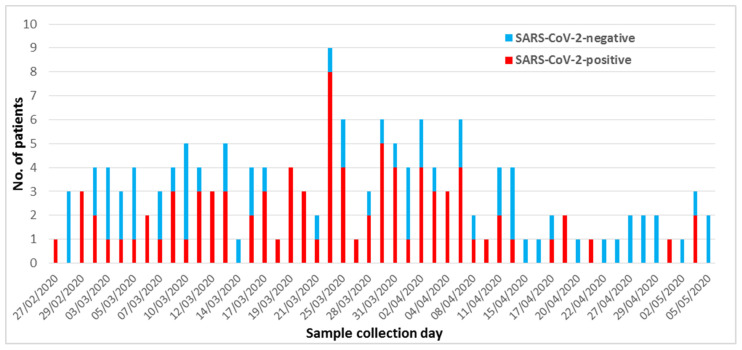
Temporal distribution of SARS-CoV-2 results among the 154 patients admitted in intensive care units.

**Figure 2 microorganisms-09-00185-f002:**
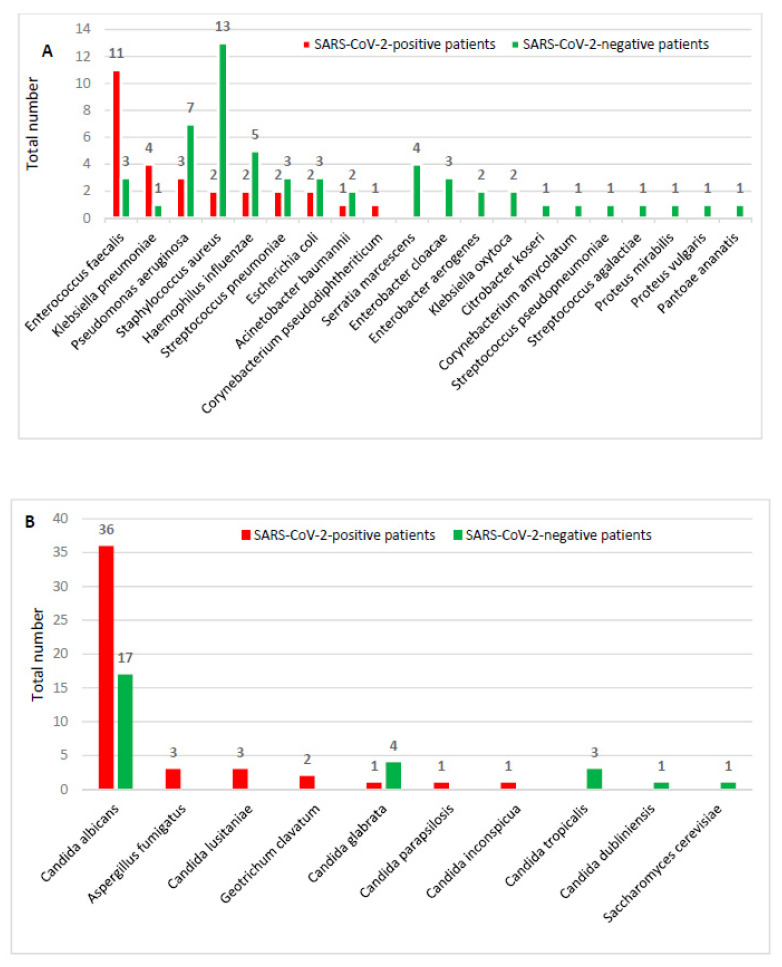
Distribution of bacteria (**A**) and fungi (**B**) among SARS-CoV-2-positive and -negative patients.

**Table 1 microorganisms-09-00185-t001:** Infectious agents detected in the overall 301 lower respiratory tract samples.

Patients (No.)	No Other Agent (No.)	Presence of		
		Bacteria (No.)	Bacteria and Fungi (No.)	Fungi (No.)
**SARS-CoV-2-positive (90)**	13	18	38	21
**SARS-CoV-2-negative (64)**	16	23	16	9

**Table 2 microorganisms-09-00185-t002:** Infectious agents detected in the first 154 lower respiratory tract samples.

Patients (No.)	No Other Agent (No.)	Presence of		
		Bacteria (No.)	Bacteria and Fungi (No.)	Fungi (No.)
**SARS-CoV-2-positive (90)**	36	10	12	32
**SARS-CoV-2-negative (64)**	19	21	14	10

## Data Availability

The data presented in this study are available in the manuscript and in the Appendix A.

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
