# Peer review of "Detection of SARS-CoV-2 and Other Infectious Agents in Lower Respiratory Tract Samples Belonging to Patients Admitted to Intensive Care Units of a Tertiary-Care Hospital, Located in an Epidemic Area, during the Italian Lockdown"

_microorganisms, 2021, doi:10.3390/microorganisms9010185_

Round 1

Reviewer 1 Report

The author's manuscript covers a hot topic, albeit only as an observational report.

It is adequately written, and very easy to follow. 

In my opinion is suitable for publication in the present form.

Author Response

We thank the Reviewer for the appreciation of our work.

Reviewer 2 Report

The work presented here describes the analysis performed on a large group of samples in respect to the presence of sars-cov-2 virus and other pathogenic organisms including bacteria, fungi and viruses.

The work is enough novel and gave and interesting overview of the potential association between sars-cov-2 infections and other ones. I don't have any corrections to suggest apart the recommendation to improve the conclusions on the discussion: is not cristal clear the role of other infections on the covid disease course, if any.

Author Response

We thank the Reviewer for the appreciation of our work and for her/his effort in suggesting to improve the conclusion in order to improve the manuscript. The conclusion section was modified accordingly (Page 9 lines 362-363): “However, the role of fungal and bacterial infections in COVID-19 disease course would require further investigation.”

Reviewer 3 Report

  1. Please add a paragraph in the “discussion” section to compare the correlation or difference of different therapies among the SARS-CoV-2 positive or negative patients infected with infectious agents including Candida albicans, Enterococcus faecalis and Staphylococcus aureus.
  2. Please add a “conclusion” section to provide the significance and achievement of this study and have an echo to the abstract.

Author Response

  1. Thanks to the Reviewer’s suggestion. Although the analysis of therapies was not the aim of our study, the data available on this information were reported in the discussion section of the revised manuscript (Page 9 lines 325-332): “Limited to the patients for whom the information on the use of the antibiotic therapy at the time of the first sample collection was available, the SARS-CoV-2 positives were prevalently treated with azithromycin and ceftriaxone mainly in combination (25 cases) followed by piperacillin-tazobactam (18 cases). Among the SARS-CoV-2 negatives, azithromycin was never indicated either in combination or alone, and several different antibiotics, alone or in combination, mainly involving penicillins with β-lactamase inhibitor (14 cases) and carbapenems (9 cases) were reported. Antifungal therapy was in use in one SARS-CoV-2-positive patient and in 4 SARS-CoV-2 negatives.

  1. According to the reviewer’s suggestion, a conclusion paragraph was added (Page 9 lines 350-363): “The data reported in this study suggest that SARS-CoV-2-positive patients admitted to ICU were are prevalently men, aged from 50 to 70 years, and that the higher mortality rate observed among SARS-CoV-2-positive patients, in comparison to SARS-CoV-2-negative ones, was is mainly attributable to older patients regardless of sex. This study highlights the different patterns in the distribution of infectious agents between the two patient categories: fungi were prevalently involved among SARS-CoV-2-positive patients. The different pattern of infectious agents, involving prevalently fungi among the SARS-CoV-2-positive patients and bacteria among the SARS-CoV-2-negative ones. Moreover, between the two patient categories also a different distribution of bacteria was observed (E. faecalis vs S. aureus) could have been influenced by the different therapies and the length of the ICU stay could have influenced this different pattern of infectious agents. However, the role of fungal and bacterial infections in COVID-19 disease course would require further investigation.”            Moreover, the abstract section was modified accordingly (Page 1 lines 28-34): “This study highlights the different patterns of infectious agents between the two patient categories: fungi were prevalently involved among SARS-CoV-2-positive patients and bacteria among the SARS-CoV-2-negative ones. The different therapies and the length of the ICU stay could have influenced this different pattern of infectious agents.”

Round 2

Reviewer 3 Report

The manuscript has been significantly improved and should be accept for publication.